# Understanding FBA Solutions under Multiple Nutrient Limitations

**DOI:** 10.3390/metabo11050257

**Published:** 2021-04-21

**Authors:** Eunice van Pelt-KleinJan, Daan H. de Groot, Bas Teusink

**Affiliations:** 1TiFN, P.O. Box 557, NL6700AN Wageningen, The Netherlands; e.van.pelt-kleinjan@vu.nl; 2Systems Biology Lab, Amsterdam Institute of Molecular and Life Sciences, Vrije Universiteit Amsterdam, 1081HZ Amsterdam, The Netherlands; daan.degroot@unibas.ch

**Keywords:** flux balance analysis, elementary flux modes, elementary conversion modes, genome-scale modeling, stoichiometric modeling, phenotype phase plane analysis

## Abstract

Genome-scale stoichiometric modeling methods, in particular Flux Balance Analysis (FBA) and variations thereof, are widely used to investigate cell metabolism and to optimize biotechnological processes. Given (1) a metabolic network, which can be reconstructed from an organism’s genome sequence, and (2) constraints on reaction rates, which may be based on measured nutrient uptake rates, FBA predicts which reactions maximize an objective flux, usually the production of cell components. Although FBA solutions may accurately predict the metabolic behavior of a cell, the actual flux predictions are often hard to interpret. This is especially the case for conditions with many constraints, such as for organisms growing in rich nutrient environments: it remains unclear why a certain solution was optimal. Here, we rationalize FBA solutions by explaining for which properties the optimal combination of metabolic strategies is selected. We provide a graphical formalism in which the selection of solutions can be visualized; we illustrate how this perspective provides a glimpse of the logic that underlies genome-scale modeling by applying our formalism to models of various sizes.

## 1. Introduction

Flux Balance Analysis (FBA) is a constraint-based optimization method that is used to study metabolic networks in steady state [1,2]. In the most basic FBA, the rate of biomass synthesis is maximized to model metabolic behavior under selection for fast growth. This biomass synthesis is limited by a constraint on some nutrient uptake rate, mimicking the limited availability of this nutrient.

However, Flux Balance Analyses often become much more complicated. For example, because the studied cell can use many different nutrients, in which case there may be many uptake rates with an upper bound, or the modeler may be interested in inferring the full metabolic behavior from the measurement of some reaction rates, which would lead to equality constraints that fix these experimentally measured rates. Alternatively, models may include a growth-independent maintenance parameter via a lower bound on the production of the energy-equivalent ATP. All these cases lead to Flux Balance Analyses with multiple constraints, and a small sample of published models indeed shows that multi-constraint models are abundant (Table 1). When FBA is used with several such constraints, it can become extremely challenging to understand why a certain solution is optimal. This severely obstructs a deep understanding of the model results, and with that, the understanding of the cell’s metabolism. It is therefore important to lay bare the logic that underlies FBA solutions. This is what we intend to do in this paper.

A comprehensive list of minimal reaction sets that can sustain a steady-state flux through the network is given by the Elementary Flux Modes (EFMs) [3]. EFMs can be viewed as the minimal metabolic strategies that a cell has at its disposal, and each metabolic state is a combination of these EFMs. Therefore, understanding FBA solutions comes down to rationalizing why the selected EFMs are used. For example, if only one flux constraint is used, often the uptake rate of the sole energy or carbon source, FBA will always select the EFM with the highest yield, i.e., the pathway that produces the most moles of biomass per mole of the limiting substrate [4,5]. Therefore, although FBA appears to explicitly maximize the biomass production rate as a proxy for maximizing the growth rate, the resultant high-yield strategy does not necessarily lead to the fastest biomass production in real organisms [4,6,7,8], because they may be limited by thermodynamic and enzyme-kinetic limitations [9,10]. Indeed, many microorganisms, and even cancer cells, do not use the high-yield pathways in fast-growth conditions, but rather use seemingly wasteful pathways that, for example, lead to ‘overflow metabolism’ [11,12,13,14,15].

Although it has been well-understood that FBA with one constraint selects the maximal-yield EFM, this logic underlying the solution cannot be readily extended to cases with multiple constraints. We will use small toy models to prove that the highest-yield pathways are not always selected in such cases. Instead, we will show that FBA selects Elementary Flux Modes based on a weighted combination of their product yields for the various constrained uptake rates.

To visualize all possible product yields for a model, we use the enumeration of *Elementary Conversion Modes* (ECMs) [22,23]. The ECMs are the minimal stoichiometric relations that the cell can use to convert substrates into products and biomass, and all ECMs together comprise all its conversion capabilities. The set of ECMs therefore span all possible overall reactions from nutrients to products, and, importantly, each Elementary Conversion Mode has a corresponding Elementary Flux Mode that leads to the same conversion. Thus, if we can decide which ECMs are selected, we also know which EFMs are selected. It is beneficial to use ECMs rather than EFMs because they can be enumerated on much larger metabolic networks [22]. We will combine the enumerated ECMs with a graphical *cost vector formalism* [24] to obtain a figure that shows exactly the part of the metabolic capabilities of the model that is relevant for the specific FBA. Aided by this graphical formalism, it can be easily seen why a certain combination of minimal metabolic pathways is optimal.

We provide several examples that illustrate the benefit of analyzing why a certain FBA solution was selected. We use the cost vector formalism to show for a genome-scale model of *Escherichia coli* how much the FBA-solution changes when one gradually changes a constraint, as such providing insight in the robustness of the FBA-predictions (known as robustness analysis [25]). Genome-scale models of *Lactococcus lactis* and *Lactobacillus plantarum* are used to illustrate how our method facilitates a deeper understanding of FBA-results, even when many constraints are used to model nutrient-rich environments.

## 2. Results

### 2.1. Small Models Show That FBA Solutions Are Not Just Selected for Maximal Yield

In this section, we will use three small networks to identify the properties of metabolic pathways that determine whether the pathway will be used in the FBA-solution. A general Flux Balance Analysis problem can be written as
(1)$$\max_{v}\left\{v_{\mathrm{BM}}\;|\;Nv=0,\;v_{\mathrm{irrev}}\geq 0,\;v_{i_{1}}\leq C_{i_{1}},\ldots ,v_{i_{K}}\leq C_{i_{K}}\right\},$$
where *N* is the stoichiometry matrix that indicates which metabolites (rows) are used or produced in each reaction of the metabolic network (columns). It is multiplied by the flux vector, v, containing all reaction rates. Some of the reaction rates also appear separately: $v_{i_{1}}$ denotes the first of *K* constrained uptake rates. Although in Equation (Equation 1) we have chosen to use only upper bounds, the results in this paper remain valid when these constraints are lower- or equality bounds. The objective reaction rate is denoted by $v_{\mathrm{BM}}$: a virtual reaction combining all cell components in the right proportions for cell-synthesis. Finally, $v_{\mathrm{irrev}}\geq 0$ indicates that all irreversible reactions must have a non-negative rate.

Since uptake fluxes are often constrained, the ratio of product formation over substrate uptake plays an important role in FBA. This ratio is called the *product-substrate yield*, $Y_{S}$. For a flux vector, v, the biomass yield on substrate *S* is given by $Y_{S}=v_{\mathrm{BM}}/v_{S}$. Note that this definition is completely general as to which product and which substrate is involved, we can thus have product-glucose yields, product-sulfur yields, and so on.

Any vector of reaction rates that satisfies the first two constraints of Equation (Equation 1) can be written as a combination of Elementary Flux Modes (EFMs) [3,26]. Since EFMs are minimal sets of reactions that can sustain a steady-state, these correspond to minimal metabolic strategies. To understand why a certain set of reactions is used in an FBA-solution, it is therefore sufficient to explain why the used EFMs were selected.

#### 2.1.1. FBA with One Constraint Selects the Maximal-Yield EFM

We here only consider *active* flux constraints, which means that $v_{\mathrm{BM}}$ could increase if the constraint was relaxed. This implies that $v_{S}=C$ in the optimal solution. By using this and the definition of product-substrate yield, we can rewrite the FBA-optimization problem as
$$\begin{array}{cc} \max_{v}\left\{v_{\mathrm{BM}}\;|\;Nv=0,\;v_{\mathrm{irrev}}\geq 0,\;v_{S}=C\right\} & =\max_{v}\left\{v_{S}Y_{S}\;|\;Nv=0,\;v_{\mathrm{irrev}}\geq 0,\;v_{S}=C\right\},\\ & =\max_{v}\left\{CY_{S}\;|\;Nv=0,\;v_{\mathrm{irrev}}\geq 0\right\},\\ & =C\max_{v}\left\{Y_{S}\;|\;Nv=0,\;v_{\mathrm{irrev}}\geq 0\right\}. \end{array}$$

This confirms that the solution v is found by selecting the steady-state flux distribution with the highest yield. In Figure 1a, we show an example network with two EFMs and only one active constraint. The EFM that uses the yellow reaction has a three times higher product yield on the limiting substrate, and is therefore used in the FBA-solution.

It is thus easy to characterize solutions of Flux Balance Analyses with only one active constraint. One could, in principle, find the solution by enumerating the EFMs, and calculate the product-substrate yields for each of them. The EFM with maximal yield will give an optimal flux distribution. It may occur that several EFMs attain the maximal yield, and in that case, all combinations of these EFMs are optimal solutions, providing a mathematical basis for “flux variability” [27], see also [28].

In practice, the number of EFMs increases rapidly with network size, often rendering the above EFM-selection recipe infeasible [29]. Even though we are initially only interested in the product-substrate yields of the EFMs, EFM theory provides no way to speed up the enumeration by focusing only on that information. Elementary Conversion Modes (ECMs), however, do give that option [22,23]. The set of ECMs not only provides a full overview of all overall conversions from nutrients to products that a cell can catalyze; enumeration with the recently introduced *ecmtool* also allows focusing only on the conversions between a selected set of nutrients and products [22]. This scales ECM enumeration to much larger networks than can be handled by EFM enumeration. Although ECMs are defined in terms of changes in external metabolite concentrations while FBA-constraints and -objectives are in terms of reaction rates, the important reaction rates can always be found by coupling the production of an external metabolite to each of these reactions. We will take a two-step approach in this paper: first, we compute the ECMs and use the conversion information to understand which ECMs constitute the FBA-solution. Then, we reconstruct a set of EFMs that gives rise to these overall conversions, because EFMs can be more easily interpreted as minimal metabolic strategies. Because these minimal strategies thus have both an EFM and an ECM associated to them, we will from now on use Elementary Mode, or EM, when the distinction does not matter.

#### 2.1.2. Under Two Constraints, EMs Are Selected on Two Different Yields

Can we now extend the above recipe—selecting the Elementary Mode with the maximal yield—to problems with several active constraints? Figure 1c shows that the answer is no. Although ${\mathrm{EM}}_{2}$ produces more product per nutrient $S_{1}$, the FBA solution is a combination of ${\mathrm{EM}}_{1}$ and ${\mathrm{EM}}_{2}$. This is because the uptake of $S_{2}$ is now also actively constrained, suddenly making the product-yield per $S_{2}$ an important determining factor. We see that ${\mathrm{EM}}_{1}$ has an infinite yield on $S_{2}$, since it does not use it at all to produce *P*. So, while ${\mathrm{EM}}_{2}$ has higher yield on $S_{1}$, ${\mathrm{EM}}_{1}$ is most efficient with $S_{2}$. We can thus no longer simply say that FBA maximizes yield, since the FBA solution has a lower yield on both substrates than the individual EMs [30].

The FBA solution is also not a combination of the maximal-yield EMs with respect to the various constraints. For example, the addition of ${\mathrm{EM}}_{3}$ in Figure 1c contradicts this intuition. The maximal-yield EMs are now ${\mathrm{EM}}_{3}$ for $S_{1}$ and ${\mathrm{EM}}_{1}$ for $S_{2}$. ${\mathrm{EM}}_{2}$ is second best in its usage of both substrates. Still, it is used in the FBA solution.

The examples in panels **c** and **e** of Figure 1 differ only in the value of one of the flux bounds ($v_{2}\leq 1$ vs. $v_{2}\leq 3$), but still the selected EMs change. This is because we change ‘how limiting’ a constraint is by changing its bound, and with that the importance of the various substrate-yields change. This shows that it is not generally possible to have a rationale for the FBA-solution in terms of substrate yields only. Selection of the best combination of metabolic strategies should at least combine information about their substrate yields and the precise value of the bounds.

### 2.2. FBA Solutions Can Be Rationalized Using Cost Vectors

The above examples show that we can only understand which EMs are selected in an FBA solution by considering their product-yields on all limiting substrates, weighted by how limiting the various substrates are. It turns out that this is indeed all information that we need, and that this can be visualized with an intuitive graphical formalism. The derivation of this graphical formalism is based on a similar formalism that we developed for understanding resource-allocation optimization problems [24].

Consider a general EM with corresponding reaction rates v; EMs are defined up to a positive constant, i.e., if we multiply v with an overall positive factor, it will still represent the same EM. To facilitate comparing the costs of making biomass between EMs, we start by rescaling all EMs such that one unit biomass is made: $v\to v/v_{\mathrm{BM}}$. The resulting vector thus gives the reaction rates that are needed to produce one unit biomass via that EM. This can also be expressed in terms of the biomass-substrate yields:$$v_{S_{j}}=\frac{v_{S_{j}}}{v_{\mathrm{BM}}}v_{\mathrm{BM}}=\frac{1}{Y_{S_{j}}}v_{\mathrm{BM}}=\frac{1}{Y_{S_{j}}},$$
where we used $v_{\mathrm{BM}}=1$ in the last equality. (Note that we could scale to any other objective flux.) The next step is to write this needed reaction rate as a fraction of the available rate. This gives a cost value, $d_{j}^{i}$, for EM *i* with respect to the *j*th constraint:$$d_{j}^{i}=\frac{\frac{1}{Y_{S_{j}}^{i}}}{C_{j}}=\frac{1}{Y_{S_{j}}^{i}C_{j}}.$$We can gather these costs in a so-called *cost vector* [24] for the *i*th EM: $d^{i}={\left[\begin{array}{ccc} d_{S_{1}}^{i} & \cdots & d_{S_{K}}^{i} \end{array}\right]}^{⊺}={\left[\begin{array}{ccc} \frac{1}{Y_{S_{1}}^{i}C_{1}} & \cdots & \frac{1}{Y_{S_{K}}^{i}C_{K}} \end{array}\right]}^{⊺}$. These cost vectors can be compared in a *K*-dimensional cost vector plot (solid vectors in Figure 1b,d,f. The *j*th coordinate of the cost vectors thus shows the fraction of the *j*th constraint that is needed to produce one unit product with that EM. Cost vectors that are far from the origin thus indicate that the EM is inefficient in its substrate-usage and vice versa.

To see how the information in the cost vectors helps us understand the FBA solutions, we rephrase the FBA-optimization as: choose a combination of the rescaled EMs, $\sum_{i=1}^{M}\lambda_{i}{\mathrm{EFM}}^{i}$, such that the sum of multiples $v_{\mathrm{BM}}=\sum_{i=1}^{M}\lambda_{i}$ is maximized and all *K* constraints are satisfied: $\sum_{i=1}^{M}\lambda_{i}d_{j}^{i}\leq 1$. For the cost vectors, this is equivalent to: find the sum of cost vectors that uses the largest number of vectors and stays within the unit square (indicated by the gray dashed lines).

An important property of this type of optimization problem is that the solution is always a combination of a number of EFMs that does not exceed the number of active uptake constraints [24]. Recall that a constraint is called active if relaxing this constraint will lead to an increase in growth rate. We prove in Appendix B that this result remains valid if we use ECMs. We thus state that: *when K constraints are active, then there is an optimum with at most K EMs.* In the examples of Figure 1a,c,e, we, respectively, had 1, 2, and 2 active constraints, and indeed this was matched by the number of used EMs.

In problems with fewer than four constraints, we can even visually examine which combination of EMs is optimal [24], as we will describe below. However, even though this visual analysis was very useful in the case of resource-allocation because these problems often have few active constraints, FBA-problems may have many more constraints. Therefore, a visual optimization is often impossible, simply because we can only make 3-dimensional plots. Still, we will illustrate with various genome-scale examples that the cost vector formalism remains very useful in these multi-constraint cases: The cost vectors indicate the strengths and weaknesses of all EMs; a vector coordinate is low when the EM is very efficient with respect to that constraint.

In a case with two constraints (such as in Figure 1, Figure 2 and Figure 3), the visual optimization relies on the knowledge that at most two EMs will be used in the optimum. Therefore, we only have to compare all candidate solutions using either a single EM or a pair of EMs. In [24], we have proven that we can rank all candidates by ordering them on the diagonal of the cost vector plot (see Figure A1 for an illustration of this scoring procedure). All EMs with a cost vector below the diagonal are scored by projecting their cost vector vertically to the diagonal, all EMs with an above-diagonal cost vector are scored by projecting the cost vector horizontally. Intuitively, this projection quantifies that the production rate of a single EM is limited by its weakest point, i.e., it is limited by the constraint that it uses most inefficiently (the largest coordinate of the cost vector). All pairs of EMs are scored by connecting their cost vectors by a line, and marking the intersection with the diagonal. As such, all candidates have been scored by a projection on the diagonal of the cost vector plot. The optimal candidate is the one that led to the lowest projection on the diagonal. This procedure immediately implies that all EMs that can be selected by FBA must have cost vectors that lie on the Pareto-front between efficient usage of the constraints. This can be clearly seen in Figure 2 and Figure 3, as we will discuss shortly. When there are three active constraints, the visual optimization procedure is similar: each candidate combination of three EMs is projected to the diagonal by finding the intersection of the plane spanned by the three cost vectors and the diagonal. Eventually, the combination of EMs that gave rise to the lowest projection is the optimal one.

Although we here focused on the case where both the objectives and constraints involve only single reaction rates, cost vectors can be used whenever the objective and the constraints can be written as linear combinations of reaction rates. For example, when a combination of ATP production and biomass production is to be maximized ($\max v_{\mathrm{BM}}+v_{\mathrm{ATP}}$), or when the summed uptake of glucose and lactose is bounded ($v_{\mathrm{glc}}+v_{\mathrm{lac}}\leq C$). In Figure A2, we show that even when a ratio of fluxes is bounded ($v_{1}/v_{2}\leq C$) we can still use cost vectors by rewriting this as $v_{1}-Cv_{2}\leq 0$. The case in which a ratio of fluxes (or yield) is to be maximized ($\max v_{1}/v_{2}$) under a set of different constraints ($v_{i}\leq C$), is thoroughly analyzed in [30]. Often, such a maximal yield will be attained in a single EFM, and therefore we could find the optimal solution by just picking the EM with the highest yield. However, when the constraints prohibit the use of only this EM, the analysis needs to be extended. As far as we can see, this, however, does not fit inside our cost vector analysis.

### 2.3. Understanding FBA Solutions for Genome-Scale Models

We illustrate how Elementary Modes can be used to understand FBA solutions using several examples of increasing complexity.

#### 2.3.1. A Core *E. coli*-Model as an Introduction

We show for a core model of *E. coli* how FBA-optimization can be fully understood with a two-dimensional plot when there are only two constraints Figure 2a. We plot the costs of all EMs with respect to the uptake constraints on glucose and oxygen. Dots that are placed closer to the origin represent EMs that are more substrate-efficient. Optimal FBA solutions can therefore always be constituted from the extreme EMs, i.e., the minimal set of EMs that spans the Pareto front that quantifies the trade-off between glucose- and oxygen-efficiency. These extreme EMs can be efficiently found with the hide-method of ecmtool (an explanation is given in Section 4). Because the hide-method facilitates ECM enumeration on larger networks and still captures all potentially optimal minimal strategies, we will often use this method on larger networks.

The full understanding of FBA solutions becomes harder, but not yet impossible, when a lower bound on some reaction rate is added. For example, in Figure 2b,c we add a lower bound on the production of ATP to model a non-growth related ATP maintenance. This constraint acts as a secondary objective: The ATP-demand has to be satisfied in the cheapest possible manner, so that the most substrate supply remains available for biomass synthesis. The ranking of EMs should now thus be done based on their efficiency in producing biomass, but also on their efficiency in producing ATP. We have therefore added a plot that shows the latter efficiencies.

It remains possible to rationalize the FBA solution. Two of the EMs (blue and red) are selected, respectively, for their substrate-efficiency in making biomass and ATP. However, combining only these two EMs would leave some of the oxygen supply unexploited. Therefore, the seemingly in-efficient brown EM can take over some of the ATP and biomass production while using relatively more oxygen than the red EM.

In larger models, the number of constraints often becomes larger than two, for example, because more nutrient uptakes are constrained. In addition, the number of secondary objectives may rise, for example, because uptake rates are fixed to their experimentally observed value, so that the uptake is forced. To facilitate the analysis in these more complicated cases, we introduce a simplified way of plotting the costs in Figure 2c. The left plot shows the costs of each EM with respect to all constraints; the right plot shows the contribution of each EM to the various objectives. These plots can be combined to understand why a certain EM was selected in the optimum.

#### 2.3.2. Optimal Solutions under Glucose- and Oxygen-Limitation of a Genome-Scale *E. coli*-Model Are Determined by Few Minimal Strategies

Figure 3a shows the 10 EMs that together span all optimal solutions under glucose- and oxygen-limitation for the genome-scale *E. coli* model iJR904 [16]. Although ECM enumeration is often not possible on genome-scale networks, Clement et al. used the hide-method to only focus on conversions between glucose, oxygen, and biomass, and parallelized the enumeration to produce this set. We have taken the reported ECMs and reconstructed a corresponding Elementary Flux Mode for each of them (see Methods). This enables us to reconstruct the consumption and excretion of metabolites that were first hidden. Products that differ between the 10 EMs are shown in Figure 3b, as a function of their oxygen per glucose requirement.

Figure 3 shows that when oxygen-limitation becomes more stringent, the optimal EMs are increasingly selected for their oxygen-efficiency. Most optimal in terms of glucose yield is full respiration: since CO_2_ is the most oxidized product, the electrons from glucose all go to oxygen (Figure 3). This EM requires 1.84 oxygen per glucose. To save on oxygen per glucose, some of the CO_2_ needs to be converted into something less oxidized, and acetate is the next best candidate: compared to other fermentation pathways, acetate production yields one additional ATP after glycolysis, and the EM only needs 1.385 oxygen per glucose. Thus, this EM trades in a bit of the CO_2_, produced in the citric-acid cycle, for acetate-production to maintain the redox balance. Since this also reduces the ATP yield on glucose, more glucose flux is required to produce one unit of objective flux. A next oxygen-saving step is to replace CO_2_ by its more reduced form formate (H_2_CO_2_). Again, the ATP yield drops, and more glucose is needed to produce biomass, but only 0.596 oxygen molecules are needed per glucose molecule. A further decrease in oxygen (to 0.066 per glucose) is possible by the production of the more reduced ethanol, and, finally, growth is possible under complete anaerobiosis by producing some succinate. Some CO_2_-production is uncircumventable, as the citric-acid cycle needs to remain active for synthesis of biomass components. For the same reason, ethanol production is increased, as it oxidizes the additional electron-carrier NADH that is produced in the citric-acid cycle. Note that, even though the number of products seems to increase incrementally, we do see large shifts in product formation, most notably the replacement of almost all CO_2_-production by formate-production. Incidentally, the production of formate requires the enzyme *pyruvate formate lyase*, which is unstable in the presence of oxygen. Therefore, this reaction can only be active in the last, anaerobic, EM in real *E. coli* cells, showing that what is stoichiometrically feasible, may not always be possible due to kinetic constraints.

Figure 3 also shows how robust a certain optimal solution is to changes in the nutrient-limitations. For example, at a fixed glucose-limitation of $v_{\mathrm{glc}}\geq -10$ and all oxygen-limitations between approximately $V_{\mathrm{ox}}\geq -7$ and $V_{\mathrm{ox}}\geq -12$, the FBA solution will use a combination of the EM with oxygen/glucose requirement 1.368 and the EM with 0.596 oxygen/glucose requirement, producing CO_2_, acetate, formate, and relatively small fractions of glycolate and ethanol. However, when oxygen-limitation is less stringent, there is only a very small region in which formate is produced, but not glycolate. This predicts that it is very unlikely to observe formate production without glycolate production. These regions of qualitatively different behavior are equivalent to the phenotype phase planes that were introduced in [32]. Our ECM-based method thus enables Phenotype Phase Plane (PhPP)-analysis without having to numerically scan the parameter space. This reduces potential numerical instabilities, and makes generalization to higher-dimensional ‘phenotype phase regions’ possible, where these could be defined as uptake rate regions that lead to the same set of active EMs.

#### 2.3.3. *Lactococcus lactis* in Rich Medium: Elementary Modes-Based Analysis Can Rationalize Solutions to Many-Constraint Optimizations

Often, many constraints are used to model an organism’s metabolism, for example when the metabolic behavior of the organism is hard to predict from first principles. This could be because we have insufficient understanding of the constraints and the objective that determine this organism’s metabolic behavior. For example, not all organisms have been selected for maximal growth rates, and it is a much debated question which constraints shape metabolism the most [33]. In such cases, the goal of FBA can be shifted to deduce why the measured metabolic behavior may appear. Intuitively, the research question shifts from “What would the optimal metabolic behavior be given this set of constraints?” to “A certain metabolic behavior is observed, can we understand why, by analyzing the costs and benefits of this behavior?” In such cases, FBA can be used by fixing all measured rates, and then infer the rest of the metabolic state through growth rate optimization. This gives rise to a total of four constraint-types: (1) upper bounds on uptake rates (e.g., limited glucose-uptake), (2) lower bounds on production rates (e.g., an ATP-maintenance demand), (3) a lower bound on uptake rates (e.g., setting an uptake rate to its observed value), and (4) an upper bound on production rates. Constraint-types (2) and (3) can also be seen as secondary objectives: The FBA solution should satisfy these demands by using as little resources (the constraints) as possible. Our EM-based method can be used to understand the consequences of the various constraints, answering which constrained reaction is used by which metabolic strategy to meet which objective.

As an example, we use a genome-scale model of *Lactococcus lactis* [18]). This lactic acid bacterium has several amino acid auxotrophies [34,35], and is therefore grown in nutrient-rich media. Amino acids are not only used for protein synthesis, but are also catabolized to yield energy-equivalents [18,36]. It is therefore hard to explain from first principles why *L. lactis* limits its uptake of amino acids, even when these are available at unlimited concentrations. Moreover, at high growth rates, lactic acid is used as a fermentation product, while at lower growth rates, it uses mixed acid fermentation, which produces more ATP per consumed carbon molecule [14,37]. It is unclear why the most carbon-efficient pathway is not always used. To still investigate the metabolic behavior, we fix certain reaction rates to their measured values, obtained from [37], in particular: amino acid uptake rates are limited, and the measured fermentation products are demanded to be excreted. Because the number of relevant metabolites is now very large (16), the hide-method is no longer enough to compute all ECMs on this genome-scale model. Therefore, we have selected the subnetwork containing only reactions that are active in the FBA-solution (see Methods for an explanation). This still yielded 14792 ECMs, of which 16 were used in the FBA-solution (as expected, because the number of ECMs is bounded by the number of constraints Appendix B [24]).

A cost-benefit analysis of all active EMs in the FBA-optimum is shown in Figure 4. Panels a–d indicate which EM takes up which part of the constraints, while panel e shows to which (secondary) objective each EM contributes. When first focusing on biomass formation, we note that the main contributor to biomass formation is EM 10651 (Figure 4e), which uses—as expected—the lactic acid fermentation pathway. EM 1740 is very similar to EM 10651, but differs in the asparagine and aspartate usage (see also the full ECMs of *L. lactis* in the Appendix A). The other Elementary Modes that produce biomass do not produce lactate and differ in their relative amino acid usage, e.g., in the uptake of threonine. These differences may be of interest for industrial applications since threonine uptake has consequences for flavor formation (i.e., 2-Methylbutanal). Besides that the uptake of some components is limited, the experimentally observed uptake rates of some amino acids seem to be higher than needed, so that we must enforce these uptake rates to describe reality (red objectives in Figure 4e). For example, the forced uptake of arginine induces EM 10073 to produce ATP while ornithine is produced, which is shown to be an efficient ATP yielding conversion when glucose uptake is limiting [18]. While most of the other arginine consuming EMs use arginine in the same fraction, EM 2866 consumes a very large fraction of the arginine pool and only contributes a little to biomass formation. This gives rise to several questions: for example, would this EM be used to produce more ATP if ornithine production was not bounded? Would that not have been more efficient than currently consuming arginine with ECM 2866? If so, what inhibits *L. lactis* in doing so, or is this a solution that has just not been found by evolution? As this example shows, the Elementary Modes-based analysis facilitates the exploration of the metabolic strategies that give rise to the optimal solutions, which could lead to further computational and experimental investigations.

#### 2.3.4. *Lactobacillus plantarum*

EFM analysis of FBA solutions has been used before to understand the growth strategy of *Lactobacillus plantarum*. This analysis was done in a less systematic way, and could only be done for a model of central metabolism where ATP-production was optimized, because it was impossible to enumerate all EFMs for the genome-scale metabolic model. It is therefore interesting to see if we still find the same conclusions when we use our approach with a full-scale model and optimizing the biomass-production. In the original study, *L. plantarum* was grown on glycerol, which initially allowed only for very slow growth. After laboratory evolution, the growth rates and measured uptake fluxes turned out to be in agreement with the FBA predictions. It was surprising, however, that both in the experiments and in the FBA prediction, on glycerol, lactate was still produced rather than oxidized to acetate. EFM analysis showed that this can be rationalized by an oxygen uptake constraint: Then it is optimal to ferment glycerol to lactate instead of acetate, because this can be done without using the limiting oxygen.

Here, we redo this EM-based analysis for the full genome-scale network [19] with the objective to make biomass. Where the model of central metabolism only had constraints on glycerol, oxygen, and citrate, we now include constraints on amino acid uptakes as well. In addition, a non-growth related ATP maintenance is demanded, giving a total of 20 constraints. We found 5541 ECMs that were enumerated for the active subnetwork, and show the costs and benefits of the 20 ECMs that are active in the FBA-solution in Figure 5. We here opted to show only the costs for the constraints with largest scaled reduced costs: glycerol, oxygen, and citrate (see Figure A3 for an overview of all constraints). Figure 5d shows that only two EMs (2295 and 2987) are used to satisfy the ATP maintenance requirement. Although these consume all of the available citrate, and a large fraction of the available glycerol, they still produce lactate. Why is it not better to produce acetate and as such save some of the glycerol- and citrate-supply? The answer, confirming the conclusions drawn in [4], is that it is better to save all oxygen for biomass synthesis. The oxygen-limitation thus overrules the glycerol-limitation in this case, so that glycerol-yield is traded off for oxygen-yield. Interestingly, the original analysis still predicted that some ATP was produced with oxygen, while this is no longer predicted in our analysis. This is a consequence of using the genome-scale model: not only the usage of constraints of different EMs are important, but also their contributions to the various objectives.

## 3. Discussion

We have investigated the logic that underlies the optimality of FBA-solutions. It was already known that there is a simple rationale explaining the solutions of FBAs when only a single nutrient uptake rate is an active (limiting) constraint: FBA selects the Elementary Mode with the maximal yield on the growth-limiting nutrient. However, even though the modeling of many organisms requires multiple constraints, this rationale could not simply be extended to multi-constraint models [30]. We found that, in general, the EMs that constitute FBA solutions are selected for their production-yields with respect to all limiting uptake rates; the importance of these different yields is dependent on the values of the flux bounds. It turned out that these selection determinants could all be summarized in a cost vector formalism that allows for a visual examination of the strengths and weaknesses of all EMs, as such aiding the understanding of why a certain EM is selected by FBA or not.

We believe that this work forms a useful contribution to the field of genome-scale stoichiometric modeling. User-friendly toolboxes [39,40,41,42,43] and automated reconstruction algorithms [44,45,46] have made constraint-based metabolic models easy to run, but interpretation of the solutions is still very hard. Pitfalls like the high redundancy of optimal solutions [27,47], or high sensitivity of the solutions to user-defined constraints, remain key challenges. Hopefully, this may be overcome by providing a deeper understanding of the mathematical logic underlying the model outcomes.

In Figure 3, we showed, in one plot, all potential solutions to Flux Balance Analyses with glucose- and oxygen-limitation for a genome-scale model of *Escherichia coli*. This result extends a similar result by Carlson and Srienc to genome-scale [48,49]. In this plot, we can for each combination of a glucose- and an oxygen-constraint, look up which EMs are optimal and why. Since only 10 EMs were needed for this plot, the optimal metabolism of *E. coli* under glucose- and oxygen-limitation can be fully grasped by understanding 10 different minimal metabolic strategies. Additionally, the visual overview enables us to test how much a certain solution would change if the glucose- or oxygen-limitation would be relaxed or tightened. This may aid in examining which parts of the FBA-solution are robust predictions, and which parts are coincidental results of the specific flux bounds that were used.

In Figure 4 and Figure 5, we have shown how solutions to multi-constraint models can be analyzed using cost vectors. In the case of *L. lactis*, for example, we investigated the amino acid usage by the various EMs. Comparing experimental observations with FBA-predictions showed that some amino acids are taken up at a much higher rate than needed. Our analysis gives an explanation: amino acids like arginine and serine can be used for ATP generation to meet the ATP maintenance demand. With *L. plantarum* we confirmed with a genome-scale model that it is optimal to anaerobically ferment glycerol and citrate to lactate to produce ATP, probably to preserve the limited oxygen supply for biomass formation. Studying the cost vectors of the same EMs for different pairs of constraints thus provides a glimpse into the strengths and weaknesses of the different EMs. This can be used to shape hypotheses answering why a certain metabolic strategy is selected, which can then be further analyzed. Such hypotheses may be of guidance to experimental perturbations that enhance or weaken the use of this strategy. Unfortunately, in this multi-constraint case, we can no longer use the visual optimization procedure that makes it possible to immediately see which EMs are used in the optimal solution under two constraints. This is simply due to our inability to make high-dimensional plots, and can thus not be overcome.

Although in our examples we have considered only prokaryotes, there is no reason that our analysis of FBA-solutions cannot be generalized to more complex cells or to cases where, for example, proteome constraints are considered instead of flux constraints. However, there may be cases in which no clear objective can be defined, for example, in the constituent cells of multi-cellular organisms, in which case our approach clearly does not work. We therefore only claim that our analysis works as long as we have a metabolic network reconstruction, a linear objective function, and linear constraints.

Our analysis of multi-constraint FBA solutions in terms of Elementary Modes was largely facilitated by two simplifying steps. First, Elementary Conversion Modes were enumerated instead of Elementary Flux Modes, which facilitates the enumeration of Elementary Modes on far larger networks [22,23]. This is mostly because there are generally much fewer ECMs than EFMs as many different EFMs map to the same overall conversion. In addition, ECM enumeration can focus on the conversions between a user-defined set of external metabolites, by ‘hiding’ information about the other metabolites. This can speed up the enumeration, and decrease the eventual number of ECMs by orders of magnitude (see [22] for a more elaborate discussion of this hide-method). We exploited this by only computing the Elementary Conversions from the limiting nutrients to biomass, since this gave exactly the necessary information to decide which of the ECMs would be used in the FBA solution. There seem to be some disadvantages to ECM-enumeration as well: (1) it does not return any information on which reactions are used, and (2) it disregards information on unconstrained metabolites that are taken up or secreted. However, this information can be retrieved easily for each ECM. We implemented a small Linear Program that finds an EFM for each enumerated ECM (described in Section 4). This shows which set of reactions could lead to the ECM, and what will happen to the first disregarded external metabolites.

Second, in the models of *L. lactis* and *L. plantarum*, we only computed the Elementary Conversion Modes for the subnetwork that was active in the FBA solution. For an FBA with only few constraints, this would have left us with a very small subnetwork, since the FBA solution would have used only a few EFMs [24]. Indeed, the active subnetwork on the core model of *E. coli* comprised only two EFMs. However, since we used the models of *L. lactis* and *L. plantarum* with around 20 constraints, there were also around 20 EFMs that thus gave rise to a much more versatile active network. Surprisingly, EFM enumeration was not even possible on the resulting active subnetwork. The enumeration of ECMs while using the hide-method to ignore all unconstrained external metabolites still returned 14,792 ECMs for the *L. lactis* model and 5541 ECMs for the *L. plantarum* model. This once again illustrates the almost impermeable complexity of genome-scale models. Again, there seems to be a disadvantage to this simplification: when the active subnetwork is selected, we can no longer get the complete overview of why the selected metabolic state was better than all the alternatives. This is because there may be alternatives that are missing because they use a reaction that was not part of the active subnetwork. This is a challenge that we cannot solve at this moment, since the number of alternatives in genome-scale models with many constraints is just too large even for ECM enumeration.

The cost vectors that we used here were previously introduced for kinetic resource-allocation models. Although their translation to FBA-type problems was relatively straightforward, it still led to new insights that we did not obtain from their original application. This motivates their generalization to more types of models. We suspect that they can be used for any constraint-based model as long as Elementary Modes can be defined, and the costs of a certain mode scale proportionally to the usage of the mode. In the current case, this demand was satisfied: if the flux through an EFM increases by a factor, then the uptake of all limiting nutrients increases by the same factor. However, we have also tried to use cost vectors for Elementary Growth Modes [50], but the inherent non-linearity of self-fabrication made it impossible to use cost vectors. The cost vectors may thus not be readily applicable for self-fabrication models like Metabolism and Expression-models [51]. On the other hand, they may very well be applicable to the various extensions to FBA (for example described in [52]).

In the past decades, there have been several efforts to use an Elementary Modes-based perspective at FBA solutions. Poolman et al. [53] proposed a method that estimates how much each Elementary Flux Mode is used for a general flux distribution. Wiback et al. [54] extended this reconstruction by obtaining the lower and upper bounds of the possible usage of each Elementary Mode. However, both of these approaches made no attempt to explain *why* a certain set of EFMs was selected. There have also been various approaches where the yield values of all EFMs were plotted in a so-called *yield space* [55,56,57] to visualize the metabolic capabilities of a network. Our work may be seen as extending these approaches by focusing specifically on those yields that determine the FBA solution, and by quantifying the importance of the different yields by taking into account the flux bound value.

The usage of the cost vectors for FBA with only two constraints reproduces results that could have been obtained through Phenotypic Phase Plane (PhPP) analysis [32]. In this approach, the rates of two uptake reaction rates are varied and for each combination an FBA is run; the resulting biomass production rate is plotted in a 3D-plot. By considering the derivatives of the biomass rates with respect to the uptake reaction rates, the plane spanned by the variation of the uptake rates could be split into distinct regions with qualitatively different metabolic behavior. Similarly, in Figure 2 and Figure 3, we could identify for which combinations of the flux bounds the FBA-solution would switch to a different pair of Elementary Modes. The Phenotypic Phase Planes are thus given by those regions in the space of constraint values for which the same Elementary Modes are selected. This makes it possible to generalize PhPP to FBAs with more than two constraints. Two additional advantages of our method is that it does not suffer from numerical instabilities when two EMs are very similar since the results of ecmtool are exact (working with fractions instead of floats), and that non-optimal Elementary Modes can also be plotted, so that it becomes clear why the optimal metabolic behavior is better than the alternatives.

## 4. Materials and Methods

### 4.1. The Use of ECMs Instead of EFMs

In this paper, we have consistently used ECMs in computations, even though we discuss EFMs during the interpretation. To understand why this is acceptable, we should quickly recapitulate the definitions of the two Elementary Modes. To facilitate readability, we will not go into details here, but an elaborate explanation of the difference between ECMs and EFMs is given in [22]. We start with the steady-state flux cone:(2)$$F=\left\{v\;|\;N_{\mathrm{int}}v=0,\;v_{\mathrm{irrev}}\geq 0\right\},$$
where $N_{\mathrm{int}}$ comprises the rows of the stoichiometry matrix *N* that correspond to internal metabolites. This cone thus comprises all steady-state flux vectors. The EFMs are defined as the *elementary vectors of the flux cone*, meaning that the set of EFMs is the minimal set such that each flux vector in F can be written as a positive conformal sum of the EFMs. It is not necessary to go into the meaning of “conformal” here, but the interested reader is referred to [58].

Analogously, the Elementary Conversion Modes are defined as the *elementary vectors of the conversion cone*:(3)$$C=\left\{\dot{c}=Nv\;|\;N_{\mathrm{int}}v=0,\;v_{\mathrm{irrev}}\geq 0\right\}=NF,$$
where the last equality indicates that the conversion cone is the image of the flux cone under the stoichiometry matrix. This means that C contains all overall conversion between external metabolites that a cell can catalyze while in steady-state.

There are two facts about EFMs and ECMs that are important for this work. First, each ECM, $\dot{c}$, there exists an EFM, e, such that $\dot{c}=Ne$. In other words, for each Elementary Conversion, there is at least one Elementary Flux Mode that leads to that overall conversion. If we thus find an optimal minimal conversion, we know that we can find a corresponding minimal metabolic strategy. Second, all information about whether an EFM will be selected in the FBA solution is contained in the ECM it maps to.

The above two facts enable us to first use the ECMs to find out which overall conversions are optimal, and then reconstruct which EFMs lead to these conversions. It is true that many EFMs may lead to this one ECM, but this just reflects that there are often many optimal FBA-solutions [5,27].

As a final note, some EFMs do not map to a single ECM, but for example to a combinations of two ECMs. However, the yield-properties of such EFMs are interpolations of the two EFMs that lead to the ECMs, and therefore this intermediate EFM is never essential in constituting the FBA solutions. This is according to the intuition that everything that this EFM can do can also be done by at least one of the two other EFMs.

### 4.2. Different Levels of Detail Can Be Used to Study FBA Solutions

In Figure 2, we have illustrated that we can use different sets of Elementary Conversion Modes to understand FBA solutions. We here describe the details of the three main sets of ECMs that we have used.

#### 4.2.1. ECMs between All External Metabolites for the Full Network

For small enough networks, we can use ecmtool to enumerate all minimal overall conversions from nutrients to products and biomass. For example, this gave 689 ECMs for the e_coli_core-model. Unfortunately, even though the number of ECMs scales better with network size than the number of EFMs, the enumeration of *all* ECMs on genome-scale networks is not yet feasible. The largest model for which all ECMs were enumerated until now is the iiT341-model of *Helicobacter pylori* comprising 485 metabolites and 554 reactions [22].

#### 4.2.2. ECMs between a Subset of External Metabolites for the Full Network

To facilitate our Elementary Mode based-analysis on genome-scale networks, we can use the hide-method that was implemented in ecmtool. This method gives all stoichiometric relations between a user-defined set of external metabolites, while the information about the other metabolites is not shown (again we refer to [22] for an elaborate explanation). We no longer compute the elementary vectors of the conversion cone (Equation (Equation 3)), but of
(4)$$C_{\mathrm{shown}}=\left\{{\dot{c}}_{\mathrm{shown}}=N_{\mathrm{shown}}v\;|\;N_{\mathrm{int}}v=0,\;v_{\mathrm{irrev}}\geq 0\right\},$$
where $N_{\mathrm{shown}}$ is the matrix with only the rows of non-hidden external metabolites. Importantly, no information about the stoichiometric coupling between non-hidden metabolites is lost during this procedure. Therefore, if we hide all external metabolites of which the uptake or production is not constrained, we still get the full information that decides which overall conversions are selected in the FBA-solution. This information is sufficient to reconstruct an EFM that leads to this conversion, which also reveals the information about the hidden metabolites.

The hide-method generally leads to a smaller set of ECMs, and to faster enumeration. Logically, this effect increases with the number of external metabolites that are hidden. Therefore, this simplification step is most useful when we have few constraints, thus requiring few unhidden metabolites. For this reason, we could enumerate the ECMs on the full genome-scale iJR904-model of *E. coli* when we focused only on conversions between glucose, oxygen, and biomass (see Figure 3).

#### 4.2.3. ECMs between a Subset of External Metabolites for the Active Network

In case we want to understand FBA solutions for genome-scale networks that also have many constraints, we have to make another simplification. For the models of *L. lactis* and *L. plantarum*, we have chosen to focus only on the subnetwork of reactions that are used in the FBA solution. For this subnetwork, it then becomes possible to enumerate the ECMs and to analyze which of these ECMs are used in the optimum. In this case, we can no longer compare the optimal ECMs with all other ECMs, but the cost vector analysis remains useful to rationalize what each of the ECMs costs, and what it brings to the optimal combination. Besides this active subnetwork, one can think of other useful subnetworks that can be used. For example, we can select all reactions that can be active according to a Flux Variability Analysis [27], or one may use experimental information to decide which reactions may be active and should thus be taken into account.

### 4.3. Outline of the Code

We here give a coarse description of the code that allows for the visualization of the costs with respect to the constraints.

*Load model and set constraints.* Depending on the research question, one can leave the default constraints that are set in the model, or one can change them. This and the subsequent optimization steps are done with the cbmpy-package [39].*Perform FBA.* We initialize the model by performing an FBA. We use the minimization of the sum of absolute reaction rates as a secondary objective. This ensures that the number of Elementary Modes will indeed be bounded by the number of constraints.*Perform reduced cost analysis.* The reduced cost of a reaction can be defined as the derivative of the objective function with respect to the reaction value. The reduced costs thus capture what happens to the objective when the reaction value is increased. When a reduced cost is nonzero in the optimum, it must have hit a constraint, and these reactions are thus of interest to us. We then select only the inhomogeneous constraints, i.e., constraints with a bound that is nonzero. Now we are left with four categories of reactions: reactions can have a positive or a negative optimal rate, and the reduced cost can also be positive or negative. A positive rate with positive reduced cost indicates that the reaction is bounded from above, so this indicates a limited production. A positive rate with negative reduced cost indicates a forced production; negative rate with positive reduced cost indicates forced uptake, and negative rate with negative reduced cost means that this uptake is limited. We mark the reactions with forced production and uptake as secondary objectives, and the limited production and uptake reactions as constraints.*Coupling an external metabolite to each constraint.* Most constraints are on exchange reactions and therefore we can immediately use the exchange of the metabolite as a marker of the reaction flux when we compute ECMs. When an internal reaction rate is constrained, we use the tag-method of ecmtool to add an external metabolite that is produced in this reaction.*Optional: Simplifying the network for ECM-enumeration.* As explained above, we can use two simplifications: (1) we can choose to consider only the active network. This means that all reactions that are zero in the FBA-solution are deleted from the model. (2) We can choose to compute the ECMs only between the external metabolites related to a constraint. If we choose this option, we here find all ECMtool-indices of the metabolites that are to be hidden.*ECM enumeration*. For this step, we used ecmtool as a Python library (as opposed to running it via the command line), of which the code can be found at https://github.com/SystemsBioinformatics/ecmtool accessed on 14 April 2021. Our computations were done with a version available at 14 April 2021. It can also be installed via the Python package manager pip.*Compute costs for all ECMs with respect to each constraint normalized to each objective.* If we have $n_{\mathrm{ecm}}$ ECMs, $n_{\mathrm{obj}}$ objects, and $n_{\mathrm{cons}}$ constraints, we thus eventually get $n_{\mathrm{obj}}$ tables of dimensions $n_{\mathrm{ecm}}$ by $n_{\mathrm{cons}}$. The rows of these tables give for each ECM the fraction of the various constraints that are used for the production of one unit objective.*Calculation of the activities of the ECMs in the FBA solution.* We first compute the overall conversion (${\dot{c}}_{\mathrm{FBA}}$) that is induced by the FBA solution for the non-hidden metabolites. Then, we perform an LP that finds this conversion as a sum of ECMs: ${\dot{c}}_{\mathrm{FBA}}=\sum_{i}\lambda_{i}{\mathrm{ECM}}_{i}$. We minimize the sum of the λ’s in this LP under the constraint that $\lambda_{i}\geq 0$ for all *i*. In principle, we could also compute the α-spectrum [47] for the various ECMs to see if they could contribute to the FBA solution, but we have not done that.*For each ECM, find a corresponding EFM.* Given an ECM ${\mathrm{ECM}}_{i}$, we can reconstruct an EFM by solving another LP: choose reaction rates v such that $N_{\mathrm{shown}}v={\mathrm{ECM}}_{i}$, with $v_{i}\geq 0$, and where $v_{i}$ is minimized. The minimization of $v_{i}$ ensures that we get a minimal solution, which will be an EFM.*Optional: reconstruct the full conversion if some metabolites were hidden.* Given the EFM, ${\mathrm{EFM}}_{i}$, we can now reconstruct the full conversion by multiplying with the full stoichiometry matrix: $\dot{c}=N\;{\mathrm{EFM}}_{i}$. As such, we get for each ECM a possible conversion between all external metabolites (i.e., including the first hidden metabolites).

#### Code Availability

The full Python-code that was used to make all figures is freely available on Github at https://github.com/SystemsBioinformatics/elementary-mode-based-FBA-rationale (accessed on 14 April 2021). This repository contains example scripts that were used to generate the figures in this paper; these can be used to build upon.

## 5. Conclusions

We originally started this investigation with the question: “Does FBA maximize yield or rate?”. One of the first things that we have realized is that this is a badly-stated question for two reasons. First, we cannot speak of yield-maximization when we have more than one limiting nutrient: our toy networks (Figure 1) showed that introducing a second limiting nutrient immediately leads to different yields that are important. When discussing the ‘yield’ of a metabolic pathway, we must realize that we are merely discussing the ratio of two reaction rates, and we must thus specify which two reaction rates are considered. Second, FBA explicitly maximizes a production rate, so the question is rather trivial. A better question turned out to be: “Based on what properties does FBA select its optima?” That is what we have tried to answer here.

## Figures and Tables

**Figure 1 metabolites-11-00257-f001:**
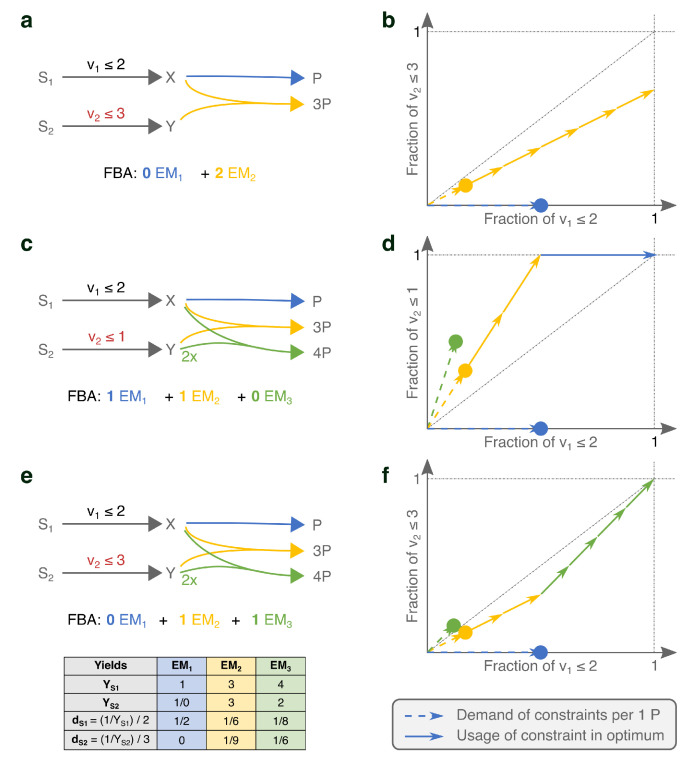
Small networks show that the product-yields on limiting substrates in combination with the values of the upper bounds determine which EMs are selected in an FBA-solution. Panels (**a**,**c**,**e**) show networks of increasing complexity and the resulting FBA solutions. In (**b**,**d**,**f**), we show the respective visualizations of the costs of the various EMs with respect to the limited nutrient supplies. The dots indicate the fractions of the constrained supplies that the corresponding EM would use for producing one *P*. The dashed vectors help to emphasize that the costs of a combination of EMs can be visualized by using vector addition (as in [24]), but for visualization purposes, these dashed vectors will not be used in subsequent figures. The solid vectors show the usage by the various EFMs of the limited supplies in the FBA solution. The computation of the vector-positions is explained in the main text, and a calculation example is shown in (**e**). (**a**,**b**) show that the maximal yield EM is selected if there is only one active constraint. In (**c**,**d**) the uptake of $S_{2}$ becomes limiting as well. Since the yellow and blue EMs are now selected, this illustrates that the product-yields on both substrates together determine which EMs are selected under two active constraints. We added the green EM to contradict that the maximal-yield EMs would always be selected. By relaxing the limit on the uptake of $S_{2}$ in (**e**,**f**), we show that the precise values of the constraint bounds influence the selection of EFMs as well.

**Figure 2 metabolites-11-00257-f002:**
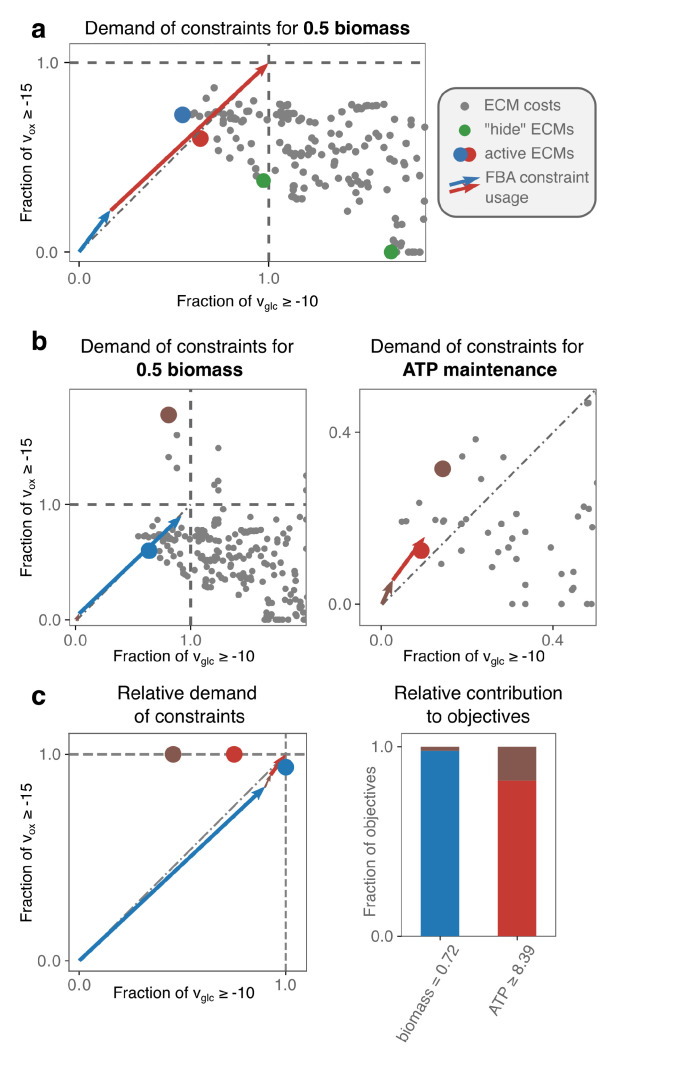
A core model of *Escherichia coli* is used to introduce our analysis of FBA solutions. We analyze the FBA solutions of the e_coli_core-model [31] with a constraint on glucose- and oxygen-uptake in (**a**), and add an ATP-maintenance demand in (**b**,**c**). In (**a**), for each of the 689 ECMs, each dot represents the fraction of the constraints that the ECM takes up when it produces biomass at a rate of 0.5. We scale to 0.5 for visualization only; it does not change the analysis. The green dots represent the costs of the ECMs that are obtained using the hide-method of ecmtool. We explain in the main text (and in [22]) that these “hide”-ECMs are extreme with respect to the chosen constraints, and can therefore always form the optimal solution. The vectors show that the red and blue EMs constitute the FBA solution. This could have been found by the visual optimization procedure as well: when all pairs of cost dots are connected by a line, the line connecting the blue and red dot intersect the diagonal closest to the origin (see text and Figure A1 for an explanation). (**b**) An ATP-maintenance demand can be imposed as a constraint, but it is better to view it as setting a secondary objective: The demanded ATP should be produced as efficiently as possible. We therefore make an additional plot that indicates the costs of the EMs for meeting this demand. Not all EMs appear in both plots, e.g., an EM that does not produce ATP will not appear in the right plot because it has ‘infinite costs’. The selection of the optimal EMs can now be understood: The red EM produces ATP most efficiently; the rest of the available glucose- and oxygen-supply can be used by the blue EM to produce biomass. However, the red and blue EM cannot exactly use all of the supply: The remainder is used by the seemingly inefficient brown EM that produces both biomass and ATP. (**c**) With more than two constraints or secondary objectives, an alternative visualization will be better suited. We only visualize optimal EMs; their dots still indicate the relative usage of the constraints, but are no longer normalized with respect to an objective. The vectors show which EM uses which fractions of the constraints. In turn, the bar plot indicates which EM contributes to which objective.

**Figure 3 metabolites-11-00257-f003:**
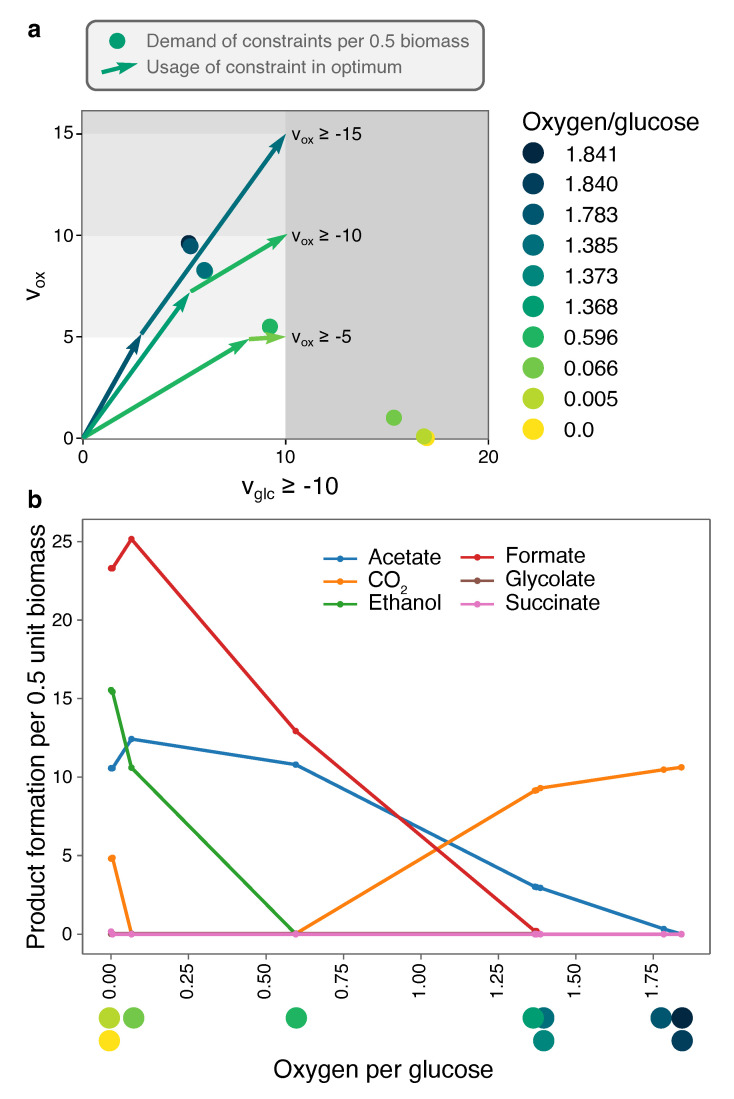
The full flexibility of *E. coli* under glucose- and oxygen-limitation can be visualized with 10 EMs. We used the set of elementary conversions between glucose, oxygen, and biomass production that are possible in a genome-scale model of *E. coli* reported in [22]. For each ECM the demand of the constraints per 0.5 unit biomass is plotted in (**a**) and characterized with its oxygen per glucose requirement. (**b**) By reconstructing for each ECM a corresponding EFM (see Methods), we could reconstruct which products are secreted by each ECM. This information is given for all shown ECMs as a function of the oxygen/glucose requirement. The three different oxygen-bounds show that under increasing oxygen-limitation, the number of different fermentation products increases. When two neighboring dots are far from each other (**a**), there is a large regime of oxygen-limitation in which the corresponding EMs are used together. Some dots are so close together that the co-utilization of the EMs is highly unlikely.

**Figure 4 metabolites-11-00257-f004:**
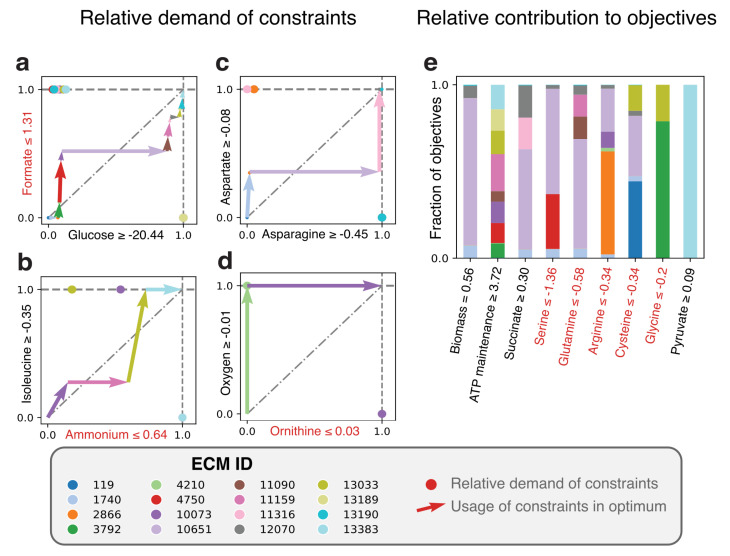
Elementary Modes-based analysis reveals the costs and benefits of all metabolic strategies that constitute the optimal solution of *Lactococcus lactis* in rich medium. ECMs were enumerated for the active network of a *L. lactis* genome-scale metabolic model [18] which was constructed based on measurements from [37]. (**a**–**d**) The costs of all the used EMs for the 8 different constraints. The pairing of the constraints in the plots is based on their scaled reduced costs, which quantifies how strongly the constraint limits the production of biomass. Glucose-limitation forms the strongest constraint, followed by formate, ammonium, etcetera. In principle, the relative costs between all pairs of constraints can be plotted. The constraints written in red are special: These force *L. lactis* to produce a certain minimal amount of product. (**e**) The contribution of each EM to the biomass objective and the secondary objectives. The secondary objectives are ordered based on their scaled reduced costs. Objectives written in red are special again: These do not demand the minimal production of something, but rather demand a certain minimal uptake rate.

**Figure 5 metabolites-11-00257-f005:**
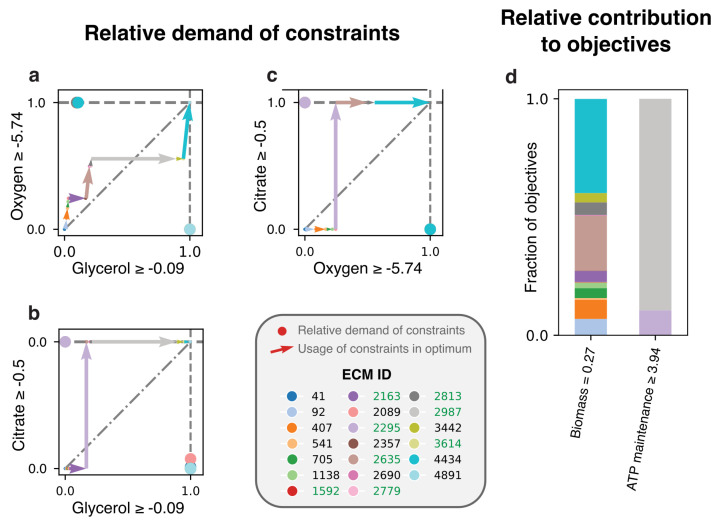
*Lactobacillus plantarum* produces ATP at lower carbon-yields to save oxygen for biomass production. The ECMs in the active subnetwork of the genome-scale *L. plantarum* model [19] were enumerated based on measured fluxes adapted from [38], and we focused on the costs of EMs for the uptake of oxygen, glycerol, and citrate. (**a**–**c**) The uptake rates of oxygen, glycerol, and citrate by each active EM are shown. To fully show the relation between the costs for these three constraints, we made a different plot for each pair. (**d**) The contribution of the EMs to the production of non-growth related ATP and to biomass production is shown. An ECM ID is written in green if the ECM produces lactate. All citrate and a large fraction of the glycerol is used by ECMs 2295 and 2987 to meet the ATP-maintenance demand. Still, this production is done without the use of oxygen and while producing lactate, even though producing ATP with oxygen and by producing acetate would have a higher carbon-yield. It thus seems that all oxygen is needed for biomass synthesis, and therefore ATP is produced via a carbon-inefficient pathway.

**Table 1 metabolites-11-00257-t001:** Published genome-scale models often have several active constraints on reaction rates. A constraint is active when it has a nonzero reduced cost.

Organism	Model	Number of Active Constraints
*E. coli*	iJR904 [16]	2
*E. coli*	iAF1260b [17]	2
*L. lactis*	[18]	10
*L. plantarum*	[19]	4
*M. tuberculosis*	iNJ661 [20]	4
*S. cerevisiae*	iMM904 [21]	3

## Data Availability

No new data were created or analyzed in this study. Data sharing is not applicable to this article.

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
