# Peer review of "Understanding FBA Solutions under Multiple Nutrient Limitations"

_metabolites, 2021, doi:10.3390/metabo11050257_

Round 1
Reviewer 1 Report
This manuscript aims to elucidate some of the parameters of Flux Balance Analysis by identifying which flux outcomes are prioritised by the metabolic network under defined conditions. It is a study which will be of interest to researchers in the area of metabolic flux analysis and ought also to interest those concerned with industrial applications where there is the necessity to predict an optimal medium to maximise production of specific products.
The manuscript provides a good scrutiny of a small area of FBA. It does remain rather limited in scope and perhaps there is not quite sufficient acknowledgement of the limitations of this approach. The authors quite rightly refer to some prokaryotic examples where metabolic pathways and cell growth conditions can be very clearly defined. They do not refer to the eukaryotic model where metabolic analysis is much more complex and where the approach of the current study displays a degree of inflexibility often due to as yet undiscovered factors which limit the simple approach of pathway utilisation based on apparent efficiency An acknowledgement of the difficulties encountered in these different models would be a good addition to the discussions. On the same note, it is not really justified to include the seminal paper by Warburg on metabolism in cancer cells among the references cited to support microbial “overflow metabolism”.
I would like to see more insight into what application this theoretical approach might hold for analysis of complex metabolic systems and what modifications would be necessary.
The manuscript is very well-written, clear and easy to follow. There is some slightly unidiomatic use of language but, in the main, this does not impair the meaning. One exception would be the sentence which begins on line 502. The meaning of this statement in unclear and I remain unsure what would be conveyed.
Reviewer 2 Report
In this work, the authors provides an EM-based graphic explanation of FBA solutions, which is an interesting extension of their previous paper. I only have a few questions:
- What if the FBA problem is added constraints of flux ratio, e.g., v1/v2 <= C? Will the addition of EM costs be effected in the plot?
- What is the story when optimizing multiply products, e.g., max(p1+p2) or max(p1/p2)?
- Is that possible the method described in Line 199-211 can be generalized to higher dimensions?
- It is interesting that the EM cost method predicts the increase of reduced products with the limitation of oxygen availability in E. coli in section 2.3.2, it will be nice to see some explanations.
